# On Unitary *t*-Designs from Relaxed Seeds

**DOI:** 10.3390/e22010092

**Published:** 2020-01-12

**Authors:** Rawad Mezher, Joe Ghalbouni, Joseph Dgheim, Damian Markham

**Affiliations:** 1Laboratoire d ’Informatique de Paris 6, CNRS, Sorbonne Université, 4 Place Jussieu, 75252 Paris CEDEX 05, France; 2Laboratoire de Physique Appliquée, Faculty of Sciences 2, Lebanese University, Fanar 90656, Lebanon; joe.ghalbouni@ul.edu.lb (J.G.); jdgheim@ul.edu.lb (J.D.)

**Keywords:** unitary *t*-design, *relaxed* seeds, approximately universal

## Abstract

The capacity to randomly pick a unitary across the whole unitary group is a powerful tool across physics and quantum information. A unitary *t*-design is designed to tackle this challenge in an efficient way, yet constructions to date rely on heavy constraints. In particular, they are composed of ensembles of unitaries which, for technical reasons, must contain inverses and whose entries are algebraic. In this work, we reduce the requirements for generating an ε-approximate unitary *t*-design. To do so, we first construct a specific *n*-qubit random quantum circuit composed of a sequence of randomly chosen 2-qubit gates, chosen from a set of unitaries which is approximately universal on U(4), yet need not contain unitaries and their inverses nor are in general composed of unitaries whose entries are algebraic; dubbed relaxed seed. We then show that this relaxed seed, when used as a basis for our construction, gives rise to an ε-approximate unitary *t*-design efficiently, where the depth of our random circuit scales as poly(n,t,log(1/ε)), thereby overcoming the two requirements which limited previous constructions. We suspect the result found here is not optimal and can be improved; particularly because the number of gates in the relaxed seeds introduced here grows with *n* and *t*. We conjecture that constant sized seeds such as those which are usually present in the literature are sufficient.

## 1. Introduction and Summary of the Results

### 1.1. Unitary t-Designs

A unitary *t*-design is an ensemble of unitaries, which, when sampled, mimic sampling from the ‘truly random’ Haar measure which chooses a unitary at random from the full continuous unitary group [1]. The usefulness of a *t*-design is that it is much simpler and more efficient to produce than sampling from the Haar measure (polynomial compared to exponential cost, respectively, [2] and [3]), yet it retains many of the useful applications. These include, but are not limited to, randomized benchmarking [4], estimating noise [5], private channels [6], photonics [7], quantum metrology [8], modeling thermalization [9], black hole physics [10], and recently demonstrations of quantum computational advantage [11,12,13].

More precisely, one can distinguish between two types of unitary *t*-designs, exact unitary *t*-designs and approximate unitary *t*-designs [14]. An exact unitary *t*-design on the *n*-qubit unitary group $U(2^{n})$ is a set of couples (we will refer to this set of couples frequently as a random unitary ensemble) ${\left\{p_{i},U_{i}\right\}}_{i=1,\ldots D}$, where *D* is a positive integer and each $U_{i}\in U\left(2^{n}\right)$ is chosen with probability $p_{i}$ ($\sum_{i=1,\ldots ,D}p_{i}=1$). An exact unitary *t*-design satisfies
(1)$$\sum_{i}p_{i}P_{\left(t,t\right)}\left(U_{i}\right)=\int_{U(2^{n})}P_{\left(t,t\right)}\left(U\right)\mu_{H}\left(dU\right),$$
where $\mu_{H}$ denotes the Haar measure on the *n*-qubit unitary group $U(2^{n})$, and $P_{\left(t,t\right)}\left(U\right)$ is any polynomial of degree exactly *t* in the matrix elements of *U*, and of degree exactly *t* in the complex conjugates of these matrix elements. It can be shown that an exact unitary *t*-design is also an exact unitary t−1 design [15] (Note that this property also holds for approximate *t*-designs). Although exact unitary *t*-designs exist for any *t* and any dimension of the unitary group [16], the search for exact unitary *t*-designs on U(d) when t>3 and d≥3 appears to be a highly nontrivial task [17]. Therefore, a natural step further is to consider a relaxation of the ’exact’ requirement and replace it with an ’approximate’ version, a so-called ε-approximate unitary *t*-design [2,14]. More explicitly, the definition of ε-approximate unitary *t*-design (or ε-approximate *t*-design for simplicity) is as follows.

**Definition** **1.**
*[2] Let*
H
*be the n-qubit Hilbert space*
${\left(C^{2}\right)}^{⊗n}$
*. A random unitary ensemble*
$\left\{p_{i},U_{i}\right\}$
*with*
$U_{i}\in U\left(2^{n}\right)$
*is said to be an ε-approximate t-design if the following holds:*
(2)$$\left(1-\varepsilon \right)\int_{U(2^{n})}U^{⊗t}\rho U^{†⊗t}\mu_{H}\left(dU\right)\leq \sum_{i}p_{i}U_{i}^{⊗t}\rho U_{i}^{†⊗t}\leq \left(1+\varepsilon \right)\int_{U(2^{n})}U^{⊗t}\rho U^{†⊗t}\mu_{H}\left(dU\right)$$
*for all*
$\rho \in B(H^{⊗t})$
*, where*
$\mu_{H}$
*denotes the Haar measure on*
$U(2^{n})$
*. For positive semidefinite matrices A and B,*
B≤A
*means*
A−B
*is positive semidefinite, ε is a positive real, and t is a positive integer (This definition is referred to as the strong definition of an ε-approximate t-design. Other definitions of ε-approximate t-designs exist, which are dependent on the application in mind, see for example [18] for an overview of these definitions.).*


Note that when ε=0, one recovers a definition of an exact unitary *t*-design which is equivalent to the definition in Equation (Equation 1) [19]. Moreover, most of the applications of exact unitary *t*-designs can be adapted to use ε-approximate unitary *t*-designs, while retaining their efficiency [5,6,8,9,11,13,14]. Finally, efficient explicit constructions of ε-approximate unitary *t*-designs for any *t* are well-established both in the circuit model [2,20] as well the measurement-based model of quantum computing [11,21,22]. For these reasons, in this work, we will focus on ε-approximate *t*-designs.

Due to the broad applications of unitary *t*-designs, one is interested in finding more efficient, and in other ways ‘better’, ε-approximate *t*-designs—for example, limiting the unitary set according to the proposed use or implementation [21]. A limiting factor in doing so is the rigid proof structure that generally follows the proof of an ε-approximate *t*-design. It is thus of high interest to be able to reduce the technical requirements involved in such a proof, which is the main topic of this work. Indeed, such technical breakthroughs will likely have application beyond *t*-designs.

### 1.2. Comparison with Previous Work

In the seminal work of [2], it was shown that n–qubit random quantum circuits composed of layers of nearest neighbor unitaries U∈U(4) drawn uniformly at random from a seed $U_{B}\subset U\left(4\right)$ (As mentioned in the abstract, a finite set of unitaries which is approximately universal in U(4) will be referred to as a seed.), sampled from an ε-approximate unitary *t*-design [14] efficiently in $poly(n,t,log\left(\frac{1}{\varepsilon }\right))$ depth. However, their proof relied on the following properties of the seed:Requirement (i): every $U\in U_{B}$ has an inverse $U^{†}\in U_{B}$.Requirement (ii): the unitaries $U\in U_{B}$ are composed entirely of algebraic entries.

The authors [2] also conjectured that the algebraic entry requirement is a technical issue (due mostly to using a result of [23]), and therefore could be dropped. Later on, in [11], it was shown that these requirements can be reduced to seeds $U_{B}$ composed partially of a seed $U_{M}$ made up of unitaries with algebraic entries and inverses in $U_{M}$; and its complement in $U_{B}$, denoted as $U_{B/M}$, which need not necessarily contain unitaries and their inverses nor be composed of algebraic entries (see also [12,20]).

In this work, we completely remove the requirements (i) and (ii) by giving examples of seeds in which every unitary in these seeds does not in general have an inverse in these seeds, nor are the unitaries in these seeds composed of algebraic entries in general, and yet converge efficiently to ε-approximate *t*-designs in a particular random circuit model which we will define explicitly below, thereby proving the conjecture proposed in [2]. We will refer to these seeds as relaxed throughout this work. However, it is to be noted that we do not mean relaxed in the sense that the unitaries making up these seeds are chosen from the Haar measure on U(4). Indeed, because our proofs are based on the partially invertible universal sets of [11], this endows the unitaries composing the relaxed seeds with some structure which makes them different from Haar distributed unitaries.

### 1.3. Main Results

The notation we will use here is the same as that in [11], but we will restate it here for the sake of using it in our proofs.

The seed $U_{B}\in U\left(4\right)$ is a partially
invertible
universal set composed of a seed $U_{M}$, which contains unitaries and their inverses, and is composed of unitaries with algebraic entries; and its complement, the seed $U_{B/M}$, which is not in general composed of unitaries and inverses, nor unitaries with algebraic entries. Define the random unitary ensemble
(3)$$B=\{\frac{1}{|U_{B}|},U_{i}\in U_{B}\}.$$

Denote the *k*-fold concatenation of *B* by
(4)$$B^{k}=\left\{\frac{1}{|U_{B^{k}}|},\prod_{j=1,\ldots k}U_{\pi (j)}\in U_{B^{k}}\right\},$$
where $U_{\pi (j)}\in U_{B}$, π is a function acting on {1,…,k} resulting in a set {π(1),…π(k)}, where $\pi \left(j\right)\in \;\{1,\ldots ,|U_{B}|\}$, the $\pi {\left(j\right)}^{\prime }s$ can be identical. There are $|U_{B}{|}^{k}$ such functions π and the *k*-fold concatenation includes all of them. $U_{B^{k}}$ is the set of all unitaries of the form $\prod_{j=1,\ldots k}U_{\pi (j)}$, with $|U_{B^{k}}\left|=\right|U_{B}{|}^{k}$. Define (This definition of $block(B^{k})$ is for even *n*, the odd *n* case follows straightforwardly.)
(5)$$block\left(B^{k}\right)=\left\{\frac{1}{|U_{B^{k}}{|}^{n-1}},\left(1_{2\times 2}⊗U_{2,3}^{j_{1}}⊗U_{4,5}^{j_{2}}⊗\ldots ⊗U_{n-2,n-1}^{j_{\frac{n}{2}-1}}⊗1_{2\times 2}\right)\left(U_{1,2}^{j_{\frac{n}{2}}}⊗U_{3,4}^{j_{\frac{n}{2}+1}}⊗\ldots ⊗U_{n-1,n}^{j_{n-1}}\right)\in U_{block(B^{k})}\right\},$$
where $U_{i,i+1}^{j}\in U_{B^{k}}$, i∈{1,…,n−1} and $j\in \{1,\ldots ,|U_{B^{k}}|\}$. Let $block^{L}\left(B^{k}\right)$ be the *L*-fold concatenation of $block(B^{k})$, defined as
(6)$$block^{L}\left(B^{k}\right)=\left\{\frac{1}{|U_{B^{k}}{|}^{(n-1)L}},\prod_{j=1,\ldots ,L}U_{\pi (j)}\in U_{block^{L}\left(B^{k}\right)}\right\},$$
where π is also as defined previously and $U_{\pi (j)}\in U_{block(B^{k})}$. Finally, let
(7)$$a=\frac{|U_{M}|}{|U_{B}|}.$$

The following theorem (Theorem 1), which holds for the above defined partially invertible universal set $U_{B}$, was one of the main results of [11], saying basically that one can obtain efficient approximate unitary *t*-designs efficiently from partially invertible universal sets in $poly\left(n,t,log\left(\frac{1}{\varepsilon^{{}^{\prime }}}\right),log\left(\frac{1}{\varepsilon_{d}}\right)\right)=O\left(n^{3}t^{12}+log\left(\frac{1}{\varepsilon^{{}^{\prime }}}\right)log\left(\frac{1}{\varepsilon_{d}}\right)\right)$.

**Theorem** **1.**
*[11] For any*
$0<\varepsilon_{d}<1$
*, and for some*
0<C<1
*, if*
(8)$$k\geq \frac{1}{log_{2}\left(\frac{1}{1+(C-1)a}\right)}\left(10t+n^{2}t-nt+n+log_{2}\left(\frac{1}{\varepsilon^{{}^{\prime }}}\right)\right)$$
*and*
(9)$$L\geq \frac{1}{log_{2}\left(\frac{1}{\varepsilon^{{}^{\prime }}+P\left(t\right)}\right)}\left(4nt+log_{2}\left(\frac{1}{\varepsilon_{d}}\right)\right),$$
*where*
(10)$$P\left(t\right)={\left(1+\frac{{\left(425{\left\lfloor log_{2}\left(4t\right)\right\rfloor }^{2}t^{5}t^{3.1/log(2)}\right)}^{-1}}{2}\right)}^{-1/3},$$
$\varepsilon^{{}^{\prime }}<1-P\left(t\right),$
*and*
$n\geq \lfloor 2.5log_{2}\left(4t\right)\rfloor$
*, then*
$block^{L}\left(B^{k}\right)$
*, formed from partially invertible universal set*
$U_{B}$
*, is an*
$\varepsilon_{d}-$
*approximate t-design on*
$U(2^{n})$
*, for any t.*


Here, ⌊.⌋ denotes the floor function. Define
(11)$$U^{k}=U_{B^{k}}-U_{M^{k}}$$
to be the seed consisting of unitaries of the form
$$U=U_{1}\ldots U_{k},$$
where for all j∈{1,…,k}, $U_{j}\in U_{B}$, and such that ∃l∈{1,…,k} and $U_{l}\in U_{B/M}$. *k* is as defined in Equation (Equation 8) in Theorem (1). $U^{k}$ in Equation (Equation 11) is the relaxed seed we will consider in this work.

We will first show that, in general, $U^{k}$ truly is relaxed by proving the following theorem, which is the first main result of this work.

**Theorem** **2.**
*For a given value of k, there is a choice of the seed*
$U_{B/M}$
*such that*
$U^{k}$
*does not verify requirement*
(ii)
*and completely violates requirement*
(i)
*.*


What is meant by completely
violates requirement (i) is that, for a choice of $U_{B/M}$, every unitary in $U^{k}$ does not have an inverse in $U^{k}$. Then, as promised, we will show that a particular random quantum circuit with seed $U^{k}$ converges to an ε-approximate *t*-design efficiently in $O(nt+log\left(\frac{1}{\varepsilon }\right))$ depth. But first, define the random unitary ensemble
(12)$$B_{1}=\left\{\frac{1}{|U^{k}|},U^{k}\right\}.$$

It is straightforward to see that
(13)$$|U^{k}|=\left(1-a^{k}\right)\left|U_{B^{k}}\right|,$$
since
(14)$$|U_{M^{k}}|=a^{k}\left|U_{B^{k}}\right|,$$
and by looking at Equation (Equation 11). $U_{M^{k}}$ is the set formed of unitaries of the form
(15)$$W=W_{1}\ldots W_{k},$$
where $W_{i}\in U_{M}$, ∀i∈{1,…,k}, and *k* is as defined in Equation (Equation 8). The random quantum circuits considered will be random unitaries in $block^{L}\left(B_{1}\right)$ defined for the random unitary ensemble $B_{1}$ (Equation (Equation 12)) in the exact same way as $block^{L}\left(B^{k}\right)$ in Equation (Equation 6) is defined for the random unitary ensemble $B^{k}$ in Equation (Equation 4), and for the exact value of *k* as in Equation (Equation 8). We will show that $block^{L}\left(B_{1}\right)$ is an ε-approximate *t*-design, first by showing that $block(B_{1})$ (This is defined for $B_{1}$ of Equation (Equation 12), in the exact same way as $block(B^{k})$ of Equation (Equation 5) is defined for the random unitary ensemble $B^{k}$ in Equation (Equation 4)) is an (η<1,t)-tensor product expander (TPE) [24,25], which is defined as follows:

**Definition** **2.**
*[24,25] A random unitary ensemble*
$\left\{p_{i},U_{i}\in U\right\}$
*is said to be an*
(η,t)
*-TPE if the following holds,*
(16)$$\|M_{t}\left[\mu \right]-M_{t}\left[\mu_{H}\right]{\|}_{\infty }\leq \eta <1,$$
*where*
$M_{t}\left[\mu_{H}\right]=\int_{U(2^{n})}U^{⊗t,t}\mu_{H}\left(dU\right)$
*,*
$M_{t}\left[\mu \right]=\sum_{i}p_{i}U_{i}^{⊗t,t}$
*, where μ is the probability measure (As shown in [26] one can shift between a probability distribution over a discrete ensemble*
$\left\{p_{i},U_{i}\right\}$
*and a continuous distribution by defining the measure*
$\mu =\sum_{i}p_{i}\delta_{U_{i}}$
*.) over the set*
U
*, which results in choosing*
$U_{i}\in U$
*with probability*
$p_{i}$
*,*
$U^{⊗t,t}=U^{⊗t}⊗U^{*⊗t}$
*, and*
$U^{*}$
*is the complex conjugate of U.*
$M_{t}\left[\mu_{H}\right]$
*and*
$M_{t}\left[\mu \right]$
*are called moment superoperators.*


Then, we will use the following proposition [20] to translate our TPE result into a result about *t*-designs

**Proposition** **1.**
*[11,20] If*
$\left\{p_{i},U_{i}\in U\right\}$
*is an*
(η<1,t)
*-TPE [24,25], then the L-fold concatenation of*
$\left\{p_{i},U_{i}\right\}$
*:*
$\left\{\prod_{j=1,\ldots ,L}p_{\pi (j)},\prod_{j=1,\ldots ,L}U_{\pi (j)}\right\}$
*is an ε-approximate t-design in the strong sense (Definition 1) when*
(17)$$L\geq \frac{1}{log_{2}\left(\frac{1}{\eta }\right)}\left(4nt+log_{2}\left(\frac{1}{\varepsilon }\right)\right).$$


π is as defined previously in Equation (Equation 4).

We now state the three theorems which establish that relaxed seeds can give rise to efficient approximate *t* designs—and are the second, third, and fourth main results of this work.

**Theorem** **3.**
$block(B_{1})$
*is an*
(η,t)−TPE
*with*
(18)$$\eta =\frac{P\left(t\right)+\varepsilon^{{}^{\prime }}}{{\left(1-a^{k}\right)}^{n-1}}+\frac{1-{\left(1-a^{k}\right)}^{n-1}}{{\left(1-a^{k}\right)}^{n-1}}.$$


Theorem (3) holds, as Theorem (1), when $n\geq \lfloor 2.5log_{2}\left(4t\right)\rfloor$ and P(t), $\varepsilon^{{}^{\prime }}$, and *k*, are exactly as defined in Theorem (1). *a* is as defined in Equation (Equation 7).

**Theorem** **4.**
*∀t, ∃*
$n_{0}\geq \left\lfloor 2.5log_{2}\left(4t\right)\right\rfloor$
*such that ∀*
n≥n0
*,*
(19)$$\frac{P\left(t\right)+\varepsilon^{{}^{\prime }}}{{\left(1-a^{k}\right)}^{n-1}}+\frac{1-{\left(1-a^{k}\right)}^{n-1}}{{\left(1-a^{k}\right)}^{n-1}}\leq 1.$$


**Theorem** **5.**
*∀t, ∃*
$n_{0}\geq \left\lfloor 2.5log_{2}\left(4t\right)\right\rfloor$
*such that ∀*
n≥n0
*,*
$block^{L}\left(B_{1}\right)$
*is an ε-approximate t-design in*
$U(2^{n})$
*in the strong sense, with L given by Equation (Equation 17), and η given by Equation (Equation 18).*


Note that Theorem (5) means, as Theorem (1), that one can obtain efficient approximate *t*-designs efficiently from relaxed seeds $U^{k}$.

The intuition behind why Theorems (3)–(5) are true is quite straightforward. $block(B^{k})$ was shown in [11] to be an (η≤1,t)-TPE [24,25]. An overwhelmingly large fraction of random unitaries (tending to one in the n,t→∞ limit, see Equation (Equation 13)) in $block(B^{k})$ are also contained in $block(B_{1})$. Therefore, one should expect $block(B_{1})$ to be an (η≤1,t)-TPE.

As a final remark in this section, note that Equations (Equation 13) and (Equation 8) tell us that the number of unitaries in the relaxed seed $U^{k}$ (Equation (Equation 11)) grows with *n* and *t*. This technical issue is due to us using the results on partially
invertible
universal sets [11] in our proofs. This is in contrast with the seeds used in [2] and [11] where these seeds were finite and were composed of a constant number of elements. We believe the results presented here are not optimal, and that finite constant-sized sets not verifying requirement (ii) and completely violating requirement (i) are sufficient to give approximate unitary *t*-designs in a random quantum circuit model efficiently in poly(n,t) depth.

### 1.4. Example: Implementation of Our Construction as a Random Quantum Circuit

In the previous subsection, we presented the main results of this work, Theorems (2)–(5), which show a mathematical construction of an ε-approximate unitary *t*-design, $block^{L}\left(B_{1}\right)$, from relaxed seeds. In practice, one can design a random quantum circuit which samples from this ε-approximate unitary *t*-design. An example of such a construction sampling from $block^{L}\left(B_{1}\right)$ is shown in Figure 1. This construction is similar to the random circuit construction in [11]. In this example, *L* is the depth of this circuit, whereas *k* controls the number of elements of the relaxed seed, which depends on the number of inputs *n* of the circuit as well as the order *t* of the design. One could also think of a translation to a measurement-based version of this random quantum circuit along the lines of work done in [11].

An important point to consider is the dependence of the circuit depth of our random circuit construction on the figure of merit *a*, defined in Equation (Equation 7). For fixed *t* and *n*, the value of η (Equation (Equation 18)) increases as *a* increases, meaning that the depth *L* of our construction increases with increasing *a*, from Equation (Equation 17). However, for large values of *t* or *n*, ${\left(1-a^{k}\right)}^{n-1}$ approaches unity, meaning that η scales asymptotically (for large *t* or *n*) as $\eta ∼P\left(t\right)+\varepsilon^{{}^{\prime }}$ (see Equation (Equation 18)). Therefore, in the limit of large *t* and *n*, the depth *L* of our random circuit construction is practically independent of *a*. (Although the value of *k* in Equation (Equation 8), which determines the cardinality of $U^{k}$, will still depend on *a*, but only up to a constant factor (see Equation (Equation 8)).) The extremal values of *a* (i.e., a=0 and a=1) are not applicable to our construction since, when a=0, the lower bound on *k* (Equation (Equation 8)) is not defined, whereas when a=1, $block(B_{1})$ is the empty set. However, it should be noted that when a=1, Theorem (1) of [11], which is the basis of the construction in this work, gives a lower bound which is in line with the lower bound on the circuit depth of the construction of approximate *t*-designs in [2] (see Theorem (1)). (The lower bound of Theorem (1) however is not as tight as that shown in [2], where the dependence on *n* in their result is linear, whereas that in Theorem (1) is cubic. Indeed, one of the open questions in [11] was whether this cubic lower bound on *n* could be reduced to a linear lower bound, which is the best one can hope to achieve for 1D random quantum circuits [2,18].)

In the next section, we present the proofs of Theorems (2)–(5).

## 2. Proofs

### 2.1. Proof of Theorem (2)

Proving requirement (ii) which is not verified by $U^{k}$ is straightforward. By our definition of the relaxed seed $U^{k}$ (Equation (Equation 11)), any unitary $U\in U^{k}$ can be written as a product of *k* unitaries in $U_{B}$ (with *k* defined in Equation (Equation 8)), $U=U_{1}\ldots U_{k}$ with at least one $U_{j}\in U_{B/M}$; and since in general $U_{B/M}$ contains unitaries with nonalgebraic entries, then the unitaries $U\in U^{k}$ are in general composed of nonalgebraic entries. To see this more clearly, let *k* be odd, and consider for example
$$U=U_{1}\ldots U_{\frac{k-1}{2}}.U_{\frac{k-1}{2}+1}\ldots U_{k-1}U_{k}\in U^{k},$$
where $U_{\frac{k-1}{2}+i}=U_{\frac{k-1}{2}-i+1}^{†}$ for $i\in \{1,\ldots ,\frac{k-1}{2}\}$ and $U_{k}\in U_{B/M}$ is a unitary with nonalgebraic entries.

Then,
$$U=U_{k}\in U^{k},$$
and is thus composed of nonalgebraic entries.

We will now prove that (i) is completely violated in general by $U^{k}$, this proof will be done by contradiction. Suppose, by contradiction, that ∀ choices of $U_{B/M}$ and for a fixed choice of $U_{M}$, ∃ $U,U^{{}^{\prime }}\in U^{k}$ such that
(20)$$U^{{}^{\prime }}=U^{†}.$$

Without loss of generality, we can write
(21)$$U=\prod_{i=1,\ldots ,k}V_{i}^{m_{i}}W_{i}^{n_{i}},$$
(22)$$U^{{}^{\prime }}=\prod_{j=k+1,\ldots ,2k}V_{j}^{m_{j}}W_{j}^{n_{j}},$$
where $V_{i},V_{j}\in U_{B/M}$, and $W_{i},W_{j}\in U_{M}$ for i∈{1,…,k}; and where $m_{i},m_{j},n_{i},n_{j}\in \left\{0,1\right\}$ with $n_{i}\neq m_{i}$ and $n_{j}\neq m_{j}$, ∀i∈{1,…k}, ∀j∈{k+1,…,2k}, and such that ∃$i_{1}\in \left\{1,\ldots ,k\right\}$ and $j_{1}\in \left\{k+1,\ldots ,2k\right\}$ such that $m_{i_{1}}=m_{j_{1}}=1$. Equations (Equation 20)–(Equation 22) imply
(23)$$V_{j_{1}}=\prod_{j=j_{1}-1,\ldots ,k+1}W_{j}^{†n_{j}}V_{j}^{†m_{j}}\prod_{i=k,\ldots ,1}W_{i}^{†n_{i}}V_{i}^{†m_{i}}\prod_{j=2k,\ldots ,j_{1}+1}W_{j}^{†n_{j}}V_{j}^{†m_{j}}.$$

Now, we will prove that Equation (Equation 23) does not hold for a general choice of $U_{B/M}$, thereby establishing a contradiction. We will consider all the possible cases as follows.

**Case 1:**$V_{j}\neq V_{j_{1}}$ ∀$j\neq j_{1}$ in Equation (Equation 23).Without loss of generality, let $U_{M}=\left\{W_{1},\ldots ,W_{n}\right\}$ and $U_{B/M}=\left\{V_{1},\ldots ,V_{m}\right\}$, with m,n∈N; and let $V_{j_{1}}=V_{m}$. Fix $\left\{W_{1},\ldots ,W_{n},V_{1},\ldots ,V_{m-1}\right\}$, and list all the possible relations of the form of the right-hand side of Equation (Equation 23), where $W_{j}\in \left\{W_{1},\ldots ,W_{n}\right\}$, ∀j∈{k+1,…,2k}, and $V_{i},V_{j}\in \left\{V_{1},\ldots ,V_{m-1}\right\}$, ∀i∈{1,…,k}, $\forall j\in \{k+1,\ldots ,j_{1}-1,j_{1}+1,\ldots 2k\}$. Since there are countably many relations of the form of the right-hand side of Equation (Equation 23) (and uncountably many choices of $V_{m}$.), choose $V_{j_{1}}=V_{m}$ such that it is not equal to any of the listed relations of the right-hand side of Equation (Equation 23). Therefore, Equation (Equation 23) does not hold in general in **Case 1**.**Case 2:** ∃$j\neq j_{1}$ such that $V_{j}=V_{j_{1}}$ in Equation (Equation 23).Here, it will be convenient to rewrite Equation (Equation 23) as
(24)$$V_{j_{1}}=\prod_{i=1,\ldots ,2k-1}C_{i}^{\pi (i)}{\left(V_{j_{1}}^{†}\right)}^{1-\pi (i)},$$
where again we take that $V_{j_{1}}=V_{m}$, $C_{i}\in \left\{V_{1}^{†},\ldots ,V_{m-1}^{†},W_{1}^{†},\ldots ,W_{n}^{†}\right\}$, and $\left\{V_{1}^{†},\ldots ,V_{m-1}^{†},W_{1}^{†},\ldots ,W_{n}^{†}\right\}$ are fixed (as in **Case 1**). π(.) is a map
i={1,…,2k−1}→π(i)∈{0,1}.We consider the two following subcases**Case 2a:**π(i)=0, ∀i∈{1,…,2k−1}.Equation (Equation 24) becomes, in this case,
(25)$$V_{j_{1}}={\left(V_{j_{1}}^{†}\right)}^{2k-1}.$$Equation (Equation 25) does not hold exactly for general choices of $V_{j_{1}}=V_{m}$, since products of the form of the right-hand side of Equation (Equation 25) can only approximate$V_{j_{1}}$ up to a given precision in general [24].**Case 2b:** ∃$i_{1}$ such that $\pi (i_{1})=1$.Equation (Equation 24) can be rewritten in this case as
(26)$$C_{i_{1}}=\prod_{i=i_{1}-1,\ldots ,1}V_{j_{1}}^{1-\pi (i)}C_{i}^{†\pi (i)}V_{j_{1}}\prod_{i=2k,\ldots ,i_{1}+1}V_{j_{1}}^{1-\pi (i)}C_{i}^{†\pi (i)}.$$Since $C_{i_{1}}\in \left\{V_{1}^{†},\ldots ,V_{m-1}^{†},W_{1}^{†},\ldots ,W_{n}^{†}\right\}$, and these unitaries are fixed, Equation (Equation 26) therefore cannot hold for a general choice of $V_{j_{1}}=V_{m}$.In order to complete the proof of Theorem (2), we should show that a $V_{m}$ exists which simultaneously violates the relations imposed in **Case 1** and **Case 2**. For a given fixed integer *k* and fixed $\left\{W_{1},\ldots ,W_{n},V_{1},\ldots ,V_{m-1}\right\}$, there is only a finite number of unitaries $V_{m}$ satisfying Equation (Equation 23) in **Case 1**. Unitaries $V_{m}$ satisfying Equations (Equation 25) and (Equation 26) (**Case 2a** and **2b**) also satisfy the relation
(27)$$det(C_{i_{1}}-\prod_{i=i_{1}-1,\ldots ,1}V_{j_{1}}^{1-\pi (i)}C_{i}^{†\pi (i)}V_{j_{1}}\prod_{i=2k,\ldots ,i_{1}+1}V_{j_{1}}^{1-\pi (i)}C_{i}^{†\pi (i)})=0.$$Using the analysis of [27], the set of unitaries $V_{m}$ satisfying relations of the form Equation (Equation 27) has zero Haar measure on U(4). This follows from the fact that one can show that there is a one-to-one mapping between these (nonidentically zero) polynomial equations in the matrix elements of $V_{m}$, and the intersection (Corresponding to partitioning the determinant into real and imaginary parts, each of which can be expressed as a trigonometric function of 16 real valued angles in [0,2π] parametrizing $V_{m}$[27].) of the zero sets of two real analytic functions on $R^{16}$. Each such zero set has a Lebesgue measure zero, therefore, their intersection (which is a subset of the two) also has Lebesgue measure zero (see [27] for more details). Therefore, the set of unitaries generated by relations of the form of Equation (Equation 27) has Haar measure zero [27]. The number of possible relations of the form of Equation (Equation 27) is countable (for fixed *k* and fixed $\left\{W_{1},\ldots ,W_{n},V_{1},\ldots ,V_{m-1}\right\}$), thus the Haar measure of the set of unitaries $V_{m}$ satisfying Equations (Equation 25) or (Equation 26) is also zero, as the countable union of measure zero sets is also measure zero. This means that we can choose $V_{m}$ to be outside a measure zero set (which is the set of unitaries satisfying Equations (Equation 23) in **Case 1**, (Equation 25), and (Equation 26)), and we would therefore have that $V_{m}$ simultaneously violates the relations imposed by **Case 1** and **Case 2**. This completes the proof of Theorem (2).

### 2.2. Proof of Theorem (3)

Define the moment superoperators
(28)$$M_{t}\left[\mu_{block(B^{k})}\right]=\sum_{i=1,\ldots |U_{B^{k}}{|}^{n-1}}\frac{1}{|U_{B^{k}}{|}^{n-1}}U_{i}^{⊗t,t},$$
where $U_{i}\in U_{block(B^{k})}$; and
(29)$$M_{t}\left[\mu_{block(B_{1})}\right]=\sum_{i=1,\ldots |U^{k}{|}^{n-1}}\frac{1}{|U^{k}{|}^{n-1}}V_{i}^{⊗t,t},$$
where $V_{i}\in U_{block(B_{1})}$. Let
(30)$$M_{t}\left[\mu_{block(B_{2})}\right]=\sum_{i=1,\ldots |U_{block(B_{2})}|}\frac{1}{|U_{block(B_{2})}|}W_{i}^{⊗t,t},$$
where $W_{i}\in U_{block(B_{2})}$. Note that $U_{block(B_{2})}$ is the complement of $U_{block(B_{1})}$ in $U_{block(B^{k})}$. Straightforward calculation using Equation (Equation 13) leads to the following relation
(31)$$M_{t}\left[\mu_{block(B^{k})}\right]={\left(1-a^{k}\right)}^{n-1}M_{t}\left[\mu_{block(B_{1})}\right]+\left(1-{\left(1-a^{k}\right)}^{n-1}\right)M_{t}\left[\mu_{block(B_{2})}\right].$$

Recalling from [11] that $M_{t}\left[\mu_{block(B_{1})}\right]$ is an (η,t)-TPE if [24,25]
(32)$$\|M_{t}\left[\mu_{block(B_{1})}\right]-M_{t}\left[\mu_{H}\right]{\|}_{\infty }\leq \eta ,$$
where $M_{t}\left[\mu_{H}\right]=\int_{U(2^{n})}U^{⊗t,t}\mu_{H}\left(dU\right)$ and $\mu_{H}$ is the Haar measure on $U(2^{n})$; using Equation (Equation 31) and a triangle inequality for norms we get
(33)$$\begin{array}{c} \|M_{t}\left[\mu_{block(B_{1})}\right]-M_{t}\left[\mu_{H}\right]{\|}_{\infty }\leq \frac{1}{{\left(1-a^{k}\right)}^{n-1}}\|M_{t}\left[\mu_{block(B^{k})}\right]-M_{t}\left[\mu_{H}\right]{\|}_{\infty }+\\ \frac{1-{\left(1-a^{k}\right)}^{n-1}}{{\left(1-a^{k}\right)}^{n-1}}\|M_{t}\left[\mu_{block(B_{2})}\right]-M_{t}\left[\mu_{H}\right]{\|}_{\infty }. \end{array}$$

Thus, $block(B_{1})$ is an (η,t)−TPE with
(34)$$\eta =\frac{1}{{\left(1-a^{k}\right)}^{n-1}}\|M_{t}\left[\mu_{block(B^{k})}\right]-M_{t}\left[\mu_{H}\right]{\|}_{\infty }+\frac{1-{\left(1-a^{k}\right)}^{n-1}}{{\left(1-a^{k}\right)}^{n-1}}\|M_{t}\left[\mu_{block(B_{2})}\right]-M_{t}\left[\mu_{H}\right]{\|}_{\infty }.$$

From a result in [11],
(35)$$\|M_{t}\left[\mu_{block(B^{k})}\right]-M_{t}\left[\mu_{H}\right]{\|}_{\infty }\leq P\left(t\right)+\varepsilon^{{}^{\prime }},$$
where P(t) and $\varepsilon^{{}^{\prime }}$ are as defined in Theorem (1). In addition, because $U_{block(B_{2})}$ is approximately universal on $U(2^{n})$ (because it is composed of unitaries which are approximately universal on U(4)), then by a result of [26],
(36)$$|M_{t}\left[\mu_{block(B_{2})}\right]-M_{t}\left[\mu_{H}\right]{\|}_{\infty }\leq 1.$$

Replacing Equations (Equation 35) and (Equation 36) in Equation (Equation 34) allows to obtain the value of η in Theorem (3).

### 2.3. Proof of Theorem (4)

The proof of Theorem (4) will also proceed by contradiction.

Suppose ∃$t_{m}$, such that ∀$n\geq \lfloor 2.5log_{2}\left(4t\right)\rfloor$,
(37)$$\frac{P\left(t_{m}\right)+\varepsilon^{{}^{\prime }}}{{\left(1-a^{k}\right)}^{n-1}}+\frac{1-{\left(1-a^{k}\right)}^{n-1}}{{\left(1-a^{k}\right)}^{n-1}}>1.$$

Notice that
(38)$$\lim_{n\to \infty }{\left(1-a^{k}\right)}^{n-1}=1,$$
with *a* and *k* as given in Equations (Equation 8) and (Equation 7), and *t* replaced by $t_{m}$. Thus, for large enough *n*, and by using Equation (Equation 38), Equation (Equation 37) reduces to
(39)$$P\left(t_{m}\right)+\varepsilon^{{}^{\prime }}∼>1.$$

Equation (Equation 39) leads to a contradiction, since by Theorem (1), $P\left(t\right)+\varepsilon^{{}^{\prime }}\leq 1$, ∀*t*. This concludes the proof of Theorem (4).

### 2.4. Proof of Theorem (5)

The proof of Theorem (5) follows directly from applying Theorems (3) and (4) in Proposition (1).

## 3. Conclusions

In this work, we have shown that one can obtain efficient approximate unitary *t*-designs from random quantum circuits with support over families of seeds which are relaxed in the sense that any unitary in the seed need not in general have its inverse in the seed, nor are the seed unitaries composed entirely of algebraic entries. This result proves and extends the scope of a conjecture proposed in [2]. The relaxed seeds presented here have a cardinality which increases with *n* and *t* (see Equation (Equation 13)). These seeds, we believe, are not optimal, and we conjecture that relaxed seeds with a constant number of elements as in [2,11] suffice to get efficient *t*-designs.

Such relaxations have natural importance when the choice of the seed is not free for various reasons; for example, in the measurement-based approach to implementing *t*-designs [11,21,22] (see also [12,13]). There, the random selection of the unitary in the ensemble is made via a measurement—that is, relying on quantum randomness, not classical randomness. This has several potential advantages, including nonadaptivity of the setup, true randomness (which may even be beyond efficient classical randomness [28]), as well as the potential for verification [29,30] and integration to broader quantum information tasks through the graph state approach [31]. A difficulty in proofs in this approach is that the strict restrictions of previous approaches [2] heavily limited the allowed measurement-based structures. Indeed, this is what motivated previous works in this direction [11,12,22]. To this end, we expect that our relaxations will allow for more diverse constructions of *t*-designs, broadening their potential implementability and integrability into quantum information networks. Furthermore, given the natural use of graph states [32] for error correction and fault tolerance [33,34], this approach may lead to much better designs of quantum advantage tolerant to noise.

Another possible application to our result is making progress towards an inverse-free version of the Solovay–Kitaev (SK) theorem [35]. Indeed, there are already hints at relations between the SK construction and unitary *t*-designs [36] (We are grateful to Michał Oszmaniec for pointing us to this result.), and our construction is the first (to our knowledge (A work which is expected to appear shortly by Oszmaniec, Horodecki, and Sawicki also manages to remove the need for inverses and algebraic entries in the seed.)) to remove the need for inverses in the base set generating the *t*-design (see technical draft for details [11]).

## Figures and Tables

**Figure 1 entropy-22-00092-f001:**
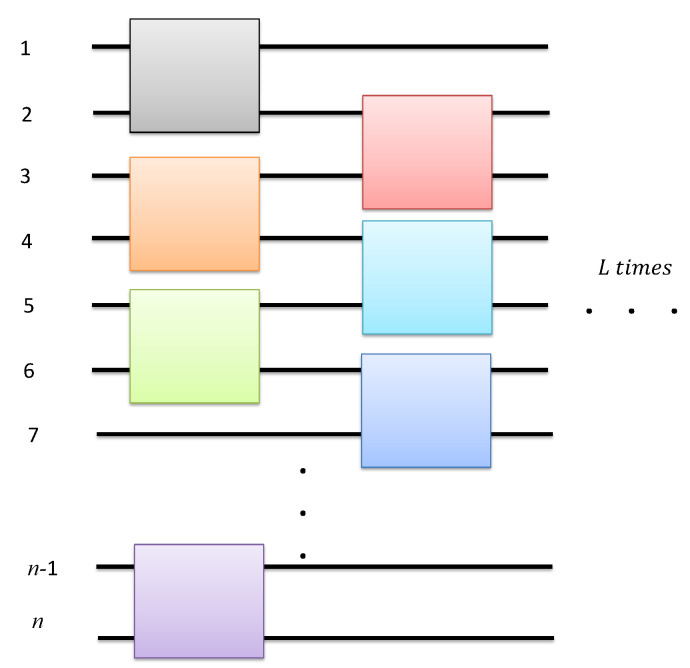
Part of the random quantum circuit sampling from the random unitary ensemble $block^{L}\left(B_{1}\right)$. The horizontal black lines numbered from 1 to *n* represent the *n* input qubits of the random quantum circuit. The colored boxes touching two horizontal lines each represent a two-qubit unitary which is chosen with uniform probability from $U^{k}$ (Equation (Equation 11)). These two-qubit unitaries act nontrivially only on the horizontal lines (qubits) they touch. The order in which these unitaries are applied is from left to right. Unitaries (boxes) aligned on the same vertical level are applied simultaneously (depth-one). The depth-two unitary shown in this figure is sampled from $block(B_{1})$. In order to sample from $block^{L}\left(B_{1}\right)$, the ε-approximate *t*-design, the random circuit shown in this figure is repeated *L* times, with *L* given by Equation (Equation 17) (see also Theorem (5)). This figure is for *n*-even, the odd *n* case follows straightforwardly.

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
