# Peer review of "On Unitary *t*-Designs from Relaxed Seeds"

_entropy, 2020, doi:10.3390/e22010092_

Round 1
Reviewer 1 Report
This paper studies the requirements for unitary t-designs, and this is a nice subject. However, the presentation of the paper is not satisfactory and the readability should be much improved.
Some definitions related to the paper (e.g. what is t-design) should be clearly given, and the motivations and background should be described in detail. The main contributions are still not clear and the abstract should list them one by one. A separate section should be placed to explain the obtained results compared to those in literature. A number of examples should be designed to illustrate the main results.Author Response
Please see the attachment

Reviewer 2 Report
The submitted manuscript discusses relaxed conditions under which approximate unitary designs can be formed. The main result is showing that two properties previously required in corresponding proofs are not needed in general.
Unitary designs are indeed commonly used in quantum information theory and more flexibel and less complex implementations are desirable. This work gives some progress in this direction and should be interesting to the field.
To the best of my judgment the results are correct. The paper is reasonably well written, apart from some minor points discussed below.
Although the results are interesting from a technical point of view, it feels like the paper could be even stronger if it would be a bit more concrete regarding implementations of the used groups in practice.
It would have also been nice if some things would have been made a bit more clear, e.g. the dependence of Equation (7) on $a$ (Figuring out whether bigger a is better or worse, requires some checking, but seems interesting to the overall understanding at that point). Also maybe state somewhere early on the practical meaning of the variables $k$ and $L$.
Finally, it would be interesting to explicitly see whether the results in [2] (or maybe even others such as [18] ) can be easily recovered from the present results.
Overall, I think this is a solid paper that extends previous results in a meaningful way. Some clarifications might be good, but afterwards I recommend the paper fur publication in Entropy.
Minor comments:
- Below line 52: "subset", not "element in"
- Generally, using . for the matrix product might not be optimal. Also sentences usually shouldn't start with math expressions. (see e.g. below Equation (29).
- Proof of Theorem 5: replacing -> applying
Round 2
Reviewer 1 Report
The paper has been improved inn a way. There is a point for improvement: the authors may mention the obtained or related results in the paper may have the potential of applications in quantum query algorithms and quantum automata, and add some references:
D.W. Qiu, S. Zheng, Generalized Deutsch-Jozsa problem and the optimal quantum algorithm, Physical Review A, 2018, 97: 062331
D.W. Qiu, G. Cai, Optimal separation in exact query complexities for Simon problem, Journal of Computer and System Sciences, 2018, 97: 83-93.
D.W. Qiu, L.Li, P. Mateus, A. Sernadas, Exponentially more concise quantum recognization of non-RMM languages, Journal of Computer and System Sciences, 2105, 81: 359-375
